# Effect of Photodynamic Therapy with the Photosensitizer Methylene Blue on Cerebral Endotheliocytes In Vitro

Vladimir I. Makarov [1,2,*], Alexey S. Skobeltsin [1,2], Anton S. Averchuk [3], Arseniy K. Berdnikov [3], Milana V. Chinenkova [3], Alla B. Salmina [3,4] and Victor B. Loschenov [1,2]

[1] Prokhorov General Physics Institute of the Russian Academy of Sciences, Moscow 119991, Russia; loschenov@mail.ru (V.B.L.)
[2] National Research Nuclear University MEPhI, Moscow 115409, Russia
[3] Laboratory of Neurobiology and Tissue Engineering, Brain Science Institute, Research Center of Neurology, Moscow 125367, Russia; antonaverchuk@yandex.ru (A.S.A.); allasalmina@mail.ru (A.B.S.)
[4] Research Institute of Molecular Medicine & Pathobiochemistry, Prof. V.F. Voino-Yasenetsky Krasnoyarsk State Medical University, Krasnoyarsk 660022, Russia
[*] Correspondence: vi.makarov@physics.msu.ru; Tel.: +7-4955038762

**Abstract:** Background: Microvessels in tumor tissue play a crucial role in meeting the metabolic needs of transformed cells, controlling the entry of xenobiotics into tumor tissue, and regulating local inflammation that promotes metastasis. Methylene blue has photosensitizing properties and can also affect dysfunctional mitochondria. Methods: The study was performed on the primary culture of CECs. The cells underwent photodynamic treatment through 660 nm laser irradiation at a power density of 300 mW/cm$^2$. The MTT, TMRE, and TUNEL assays were used to assess the survival, redox metabolism, mitochondrial activity, and apoptosis of CECs. Additionally, the metabolic activity of cells was evaluated using FLIM by measuring the fluorescence lifetime of NADH and FAD. Results: When CECs were incubated with MB, there was an increase in mitochondrial activity that was dependent on the concentration of MB. Additionally, mitochondrial activity increased when the CECs were exposed to 660 nm laser irradiation at an energy dose of up to 5 J/cm$^2$. Following PDT, a slight shift towards oxidative phosphorylation was observed. Conclusions: In vitro application of MB accumulation or laser irradiation causes a shift in the redox status of CECs towards increased reducing activity, without causing any cell damage. However, the combined action of PS and laser radiation has the opposite effect on the redox status of cells, resulting in an increase in the oxidized form of FAD.

**Keywords:** photodynamic therapy; methylene blue; FLIM; cerebral microvascular endothelium; cell metabolism assessment





## 1. Introduction

The U.S. Central Brain Tumor Registry reports that the incidence of brain tumors from 2014 to 2018 was 24.25 per 100,000, with 7.06 per 100,000 being malignant and 17.18 per 100,000 being benign. The five-year survival rate for all malignant brain tumors is gradually increasing, but remains low at approximately 35.6%, and only 6.6% for glioblastoma [1]. During surgery to remove a tumor, it is important to preserve adjacent brain structures as much as possible to minimize the risk of neurological deficits in patients post-surgery. The search for new combined methods arises from the unsatisfactory results of standard treatment for brain tumors, with the aim of improving treatment effectiveness.

Currently, researchers are developing and testing methods to increase the five-year survival rate in malignant neoplasms [2]. The methods used for cancer treatment include molecular targeting therapy, immunotherapy, oncolytic virotherapy, and photodynamic therapy (PDT) with photosensitizers.

PDT is a therapeutic technique that destroys tumor cells using reactive oxygen species (ROS) induced by light after sensitization with photosensitizers. Several photosensitizers, including porphyrins, 5-aminolevulinic acid, chlorines, and phthalocyanines, have been studied in glioblastoma over the past five years with promising results [3]. However, recent experimental applications have shown other effects that may improve overall efficacy against high-grade gliomas [4]. The clinical reports suggest that the PDT intervention may significantly stimulate the endothelial cells of the blood–brain barrier (BBB) for specific responses. Identifying and exploring the pivotal changes in endothelial cells after PDT intervention may provide necessary information to guide the clinical application of PDT on GBM therapy [5]. The study investigated the temporary opening of BBB due to PDT [6,7], initiation of an immune response against tumor cells [8–10], and deactivation of tumor-associated macrophages that promote tumor progression [11].

It is well established that during tumor progression, a local microenvironment is formed that contributes to the survival of transformed cells. Tumor-associated cells such as macrophages, lymphocytes, and stromal cells, as well as microvascular CECs, significantly contribute to the formation of this microenvironment [12]. Assessing and managing the tumor microenvironment is a new direction in diagnostics and antitumor therapy [13].

Microvessels in tumor tissue play a crucial role in meeting the metabolic needs of transformed cells, controlling the influx of xenobiotics, including drug compounds, into tumor tissue, and regulating local inflammation that promotes metastasis [14]. A correlation has been demonstrated between the intensity of neoangiogenesis, inflammation, local immune response, and the tumor's ability to metastasize in diffuse gliomas [15]. Tumor tissue induces neoangiogenesis through the release of pro-angiogenic molecules, such as VEGF and PDGF, due to localized hypoxia. This process involves the remodeling of existing capillaries, degradation of extracellular matrix, and migration and proliferation of CECs. In the case of gliomas, these steps are observed. Current approaches to inhibiting angiogenesis in gliomas have primarily centered on utilizing antagonists of proangiogenic factors or modulators of HIF-1 expression in the tissue [16]. However, these approaches have significant drawbacks and limitations [17,18], including a general lack of understanding of cerebral angiogenesis mechanisms [19].

It is important to note that the blood–brain barrier (BBB) is composed of brain vascular endothelial cells, pericytes, astrocytes, and neurons. The BBB's impermeability is due to unique cellular components, with CECs playing a major role through tight junctions (TJ) formed by transmembrane TJ proteins and other proteins from zonula occludens [20]. As a result of these structures, hydrophilic, highly charged, and large (>400 Da) molecules are unable to penetrate the barrier. This barrier regulates the movement of molecules and cells between the bloodstream and the brain, protecting the brain parenchyma from harmful agents while also excluding therapeutic agents that may interfere with treatment [21]. In brain glioblastoma, the BBB's integrity is often compromised due to tumor heterogeneity, but it retains its properties in some areas of the tumor [22]. Research has demonstrated a direct correlation between mitochondrial activity and the permeability of the BBB [23]. This means that by influencing mitochondrial activity, it may be possible to regulate BBB permeability, particularly in the context of drug delivery.

Therefore, investigating the mechanisms of the PDT effect on CECs is crucial for the advancement of tumor therapies.

Methylene blue (MB) is an FDA-approved drug used to treat cyanide poisoning, carbon monoxide poisoning, and methemoglobinemia [24]. It has been shown to attenuate pathological and neurobehavioral impairments in animal models of Alzheimer's disease, Parkinson's disease, and ischemic stroke [25]. Methylene blue (MB) is a promising photosensitizer (PS) for photodynamic therapy (PDT) of glial tumors. MB not only functions as a PS under light exposure, but also targets dysfunctional mitochondria of cells, as shown in Figure 1. MB is a highly lipophilic compound that can effectively cross the blood–brain barrier (BBB) and has a strong affinity for mitochondria. Unlike other antioxidants, such as MitoQ and MitoVitE, MB reduces free radical production by bypassing complex I/III

activity instead of removing free radicals. Studies have shown that MB can partially restore the membrane potential in mitochondria inhibited by complex III in both mice and rats. MB, acting as an electron donor, can increase brain cytochrome oxidase expression and oxygen consumption in vivo. Additionally, it has been demonstrated that small amounts of MB can prevent the inhibition of mitochondrial cytochrome c oxidase activity mediated by nitric oxide (NO). These mechanisms are described in detail in the studies of E. Klosowski et al. and Xue, H et al. [26,27]. MB promotes cytochrome c reduction, bypasses complex II, increases oxygen consumption, and produces ROS in the absence of light [28]. Thus, under different conditions, MB has a multidirectional effect on mitochondria. It has a suppressive action in hypoxia, but a stimulatory one in normoxia.

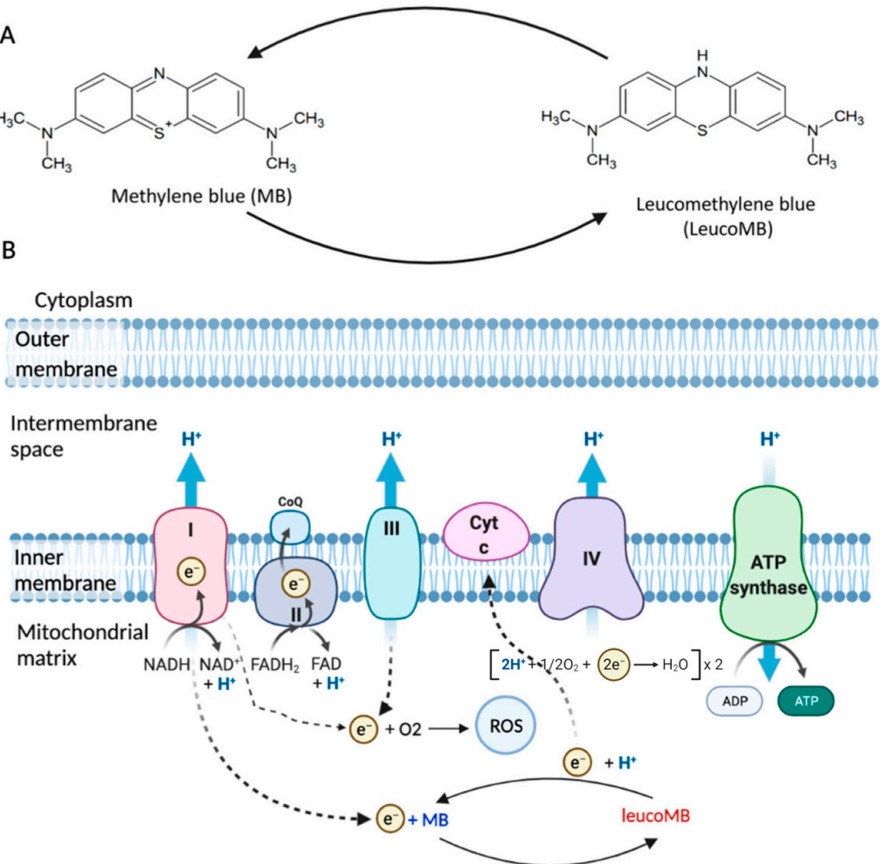

**Figure 1.** The structure and function of MB. (**A**) MB is a phenothiazine derivative, and its reduced form is leucoMB. (**B**) Mitochondrial electron transport chain (ETC) is associated with ATP production and ROS production. MB can work as a catalytic redox cycler in mitochondria and bypass complex I/III activity (figure adapted from Xue, H et al. [27]).

MB is a water-soluble compound authorized for clinical use. MB is widely used in medical practice, including oncology, dermatology, and other fields of medicine, for the treatment of various diseases. According to [29], MB increases the oxygen consumption of tumor tissues through aerobic glucose breakdown. The catalytic properties of MB against tumors are due to its interaction with lactic acid, which is produced by glycolysis. MB can reduce divalent iron in methemoglobin to a bivalent state, which corresponds to normal hemoglobin [30]. MB has also been reported to have a positive effect on peripheral blood flow [31]. However, the use of MB as a single drug or in combination with other drugs for cancer treatment is currently limited [32]. Although many properties of MB remain unexplored, they show promise in therapy.

Literature data show that the efficiency of MB in generating singlet oxygen is high and weakly dependent on solvent. The quantum yield of singlet oxygen generation ranges from

0.50 to 0.57 in methanol, 0.50 to 0.52 in ethanol and acetonitrile, and 0.50 to 0.60 in water (solutions, in air) [33,34]. However, in vitro studies have shown that these parameters are significantly reduced [35]. The quantum yield of singlet oxygen generation ($\varphi\Delta$) by MB in HeLa cells ranged from 0.0028 to 0.0065. The low values of $\varphi\Delta$ are attributed by the authors to the tendency of MB to shift from a type II photosensitization mechanism, which involves energy transfer to oxygen resulting in the formation of singlet oxygen, to a type I mechanism, which involves electron transfer resulting in the formation of MB+ semi-reduced and semi-oxidized radicals.

The objective of this work was to assess the feasibility of regulating the metabolic status of cerebral endothelial cells. This regulation directly affects the intensity of angiogenesis mechanisms and BBB permeability in the glioma microenvironment. To achieve this goal, PDT with MB was used. For the first time, the effect of MB and PDT on the regenerative ability of a cell population, mitochondrial activity, and metabolic activity of CECs was quantified. The modulation of key energy production mechanisms in mitochondrial-enriched CECs by exogenous photosensitizers can be considered as the basis for developing clinical protocols aimed at controlling BBB permeability, angiogenesis, and barrier genesis in pathological conditions associated with impaired BBB integrity and cerebral angiogenesis, such as brain tumors or chronic neurodegeneration.

## 2. Results

Figure 2a displays the extinction and fluorescence spectra of MB in the culture medium (DMEM-F12, 20% fetal calf serum, glutamine, and glucose) at concentrations of 1 and 10 mg/L (3 and 30 µM, respectively). It is evident that the culture medium exhibits a strong absorption peak at a wavelength of 556 nm. The difference in absorbance intensity is attributed to varying concentrations of the culture medium (DMEM) in the sample, resulting from the dilution of the DMEM stock solution with MB aqueous solution. Both samples containing MB exhibit an absorption peak with a maximum at 660 nm. However, a second intense absorption peak was observed in the red region of the spectrum at 600–630 nm for an MB concentration of 10 mg/L, indicating the presence of a high concentration of dimers in this sample.

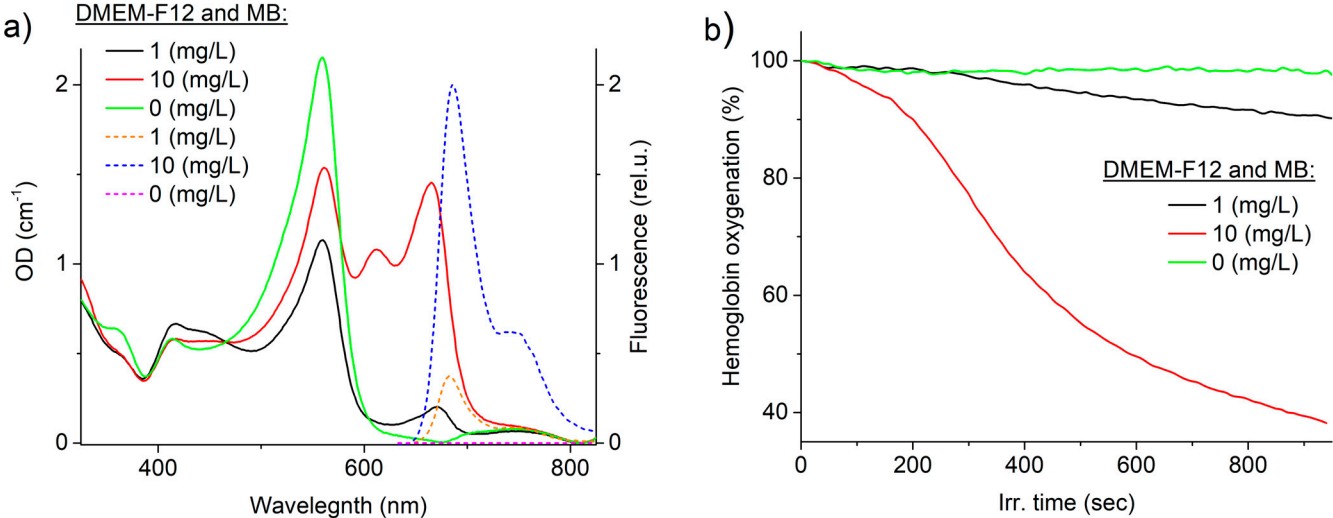

**Figure 2.** (**a**) Optical density (per cm$^{-1}$; solid lines) and fluorescence spectra (dashed lines) of 1 and 10 mg/L MB solution in culture medium (DMEM-F12, 20% fetal calf serum, glutamine, and glucose); (**b**) dependence of hemoglobin oxygenation change in the sample on laser exposure time in samples with 0, 1, and 10 mg/L MB concentrations.

Fluorescence was excited using laser radiation with a wavelength of 632.8 nm due to the high absorption of the culture medium in the 400–600 nm spectral region. A shift

of approximately 3 nm towards longer wavelengths in the fluorescence maximum and an 8 nm broadening of the peak at half-height were observed at higher concentrations, indicating fluorescence reabsorption in the sample. The culture medium does not fluoresce significantly under this excitation mode.

Figure 2b depicts the relationship between hemoglobin oxygenation and laser exposure time (660 nm, 100 mW/cm$^2$) in samples with 1 and 10 mg/L MB concentration, as well as in samples without MB. The measurement method used cannot determine the quantum yield of $^1O_2$ generation. However, it can compare the oxygen-dependent photodynamic efficiency of PSs. Based on the obtained data, it can be concluded that MB works through the oxygen mechanism under laser exposure, as the oxygen level in samples with MB decreases.

It is important to note the shape of the hemoglobin deoxygenation curve, which typically exhibits an exponential decline in most PSs such as protoporphyrin IX, derivatives of aluminum phthalocyanine, and chlorine e6. In this study, a sigmoidal curve indicating a decrease in oxygen content was observed for both 1 mg/L and 10 mg/L of MB. This phenomenon may be attributed to the dissolution of dimers during local heating under laser exposure. Two competing processes were observed: first, MB photobleaching, and second, an increase in the concentration of MB monomers. During the initial stage of laser exposure, the sample contained numerous MB dimers that were unable to effectively transfer energy to singlet oxygen. However, during the process of local laser heating, the solubility of MB increased, and the rate of oxygen consumption in the system gradually rose until it reached a plateau. During the laser exposure process, the oxygen consumption rate decreased exponentially as the oxygen concentration in the sample and the MB concentration due to photobleaching decreased. The proposed mechanism lacks strict proof and requires further investigation, and it was not the main purpose of this work. However, the mechanism described above appears to be the most probable.

To investigate the effects of laser exposure with MB on CECs, MTT, TUNEL, and TMRE assays were performed either separately or in combination.

Figure 3 displays microscopic fluorescence images of CECs after 1 h of incubation with MB at a concentration of 10 mg/L. A uniform accumulation of CECs is observed in the cytoplasm, but there are significant differences between the cells.

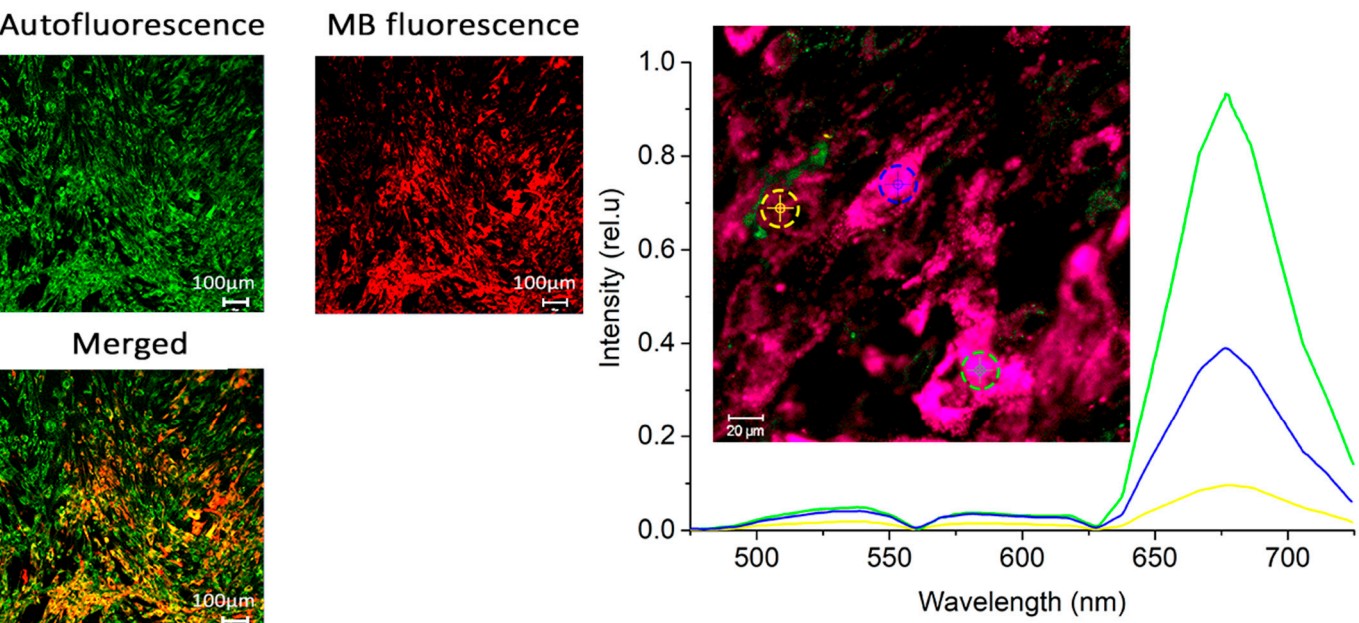

**Figure 3.** (**Left**) Microscopic fluorescence images of CECs 1 h after incubation with MB at a concentration of 10 mg/L. The green channel is autofluorescence and the red channel is MB fluorescence.

Bottom is the merged image. (**Right**) Microscopic fluorescence images of CECs and fluorescence spectra at selected image points 1 h after incubation with MB at a concentration of 10 mg/L. The color of the marker on the image corresponds to the color of the spectrum on the graph. The images were obtained by CW irradiation with a 488 nm argon laser with a wavelength of 5 mW and a 633 nm helium–neon laser with a power of 1 mW. The image was composed of 100 individual images (10 × 10), each irradiated for 7.75 s. The image in the inset above the spectrum consisted of 4 images (2 × 2) and the irradiation time for each was 30.98 s.

Figure 4 shows the integral fluorescence of MB in CECs after incubation with MB at concentrations of 1 and 10 mg/L for 1 h, as well as its photobleaching during PDT.

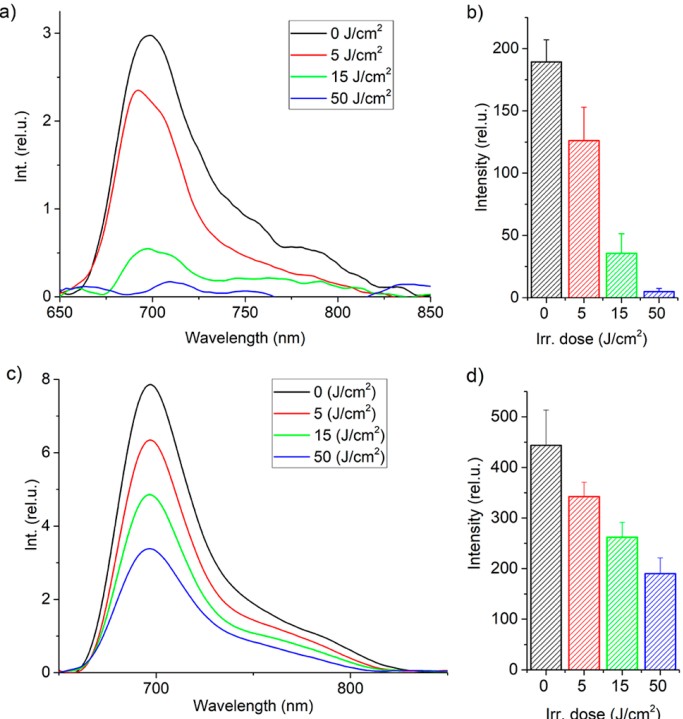

**Figure 4.** Spectra and intensity of integrated MB fluorescence in CECs in each well of the plate at MB concentration (**a**,**b**) 1 mg/L and (**c**,**d**) 10 mg/L before PDT and after irradiation at doses of 5, 15, and 50 J/cm$^2$ (660 nm, 300 mW/cm$^2$). The bars show the average values, and the error bars show the standard deviation.

A decrease in MB fluorescence intensity in CECs was observed under laser exposure due to photobleaching. Photobleaching is a complex process that involves direct photodissociation of MB molecules and destruction of MB molecules mediated by ROS and other specific reactions [36]. It is important to note that at a light dose of 15 J/cm$^2$ and a concentration of 1 mg/L, all MB was consumed, indicating that PDT-induced ROS does not affect CECs during subsequent irradiation. For a concentration of 10 mg/L, the concentration of MB inside CECs decreased by more than 50%. Figure 4 demonstrates different photobleaching rates, which may be attributed to various factors. At high concentrations of MB, the local oxygen concentration near the excited MB molecules does not have enough time to recover due to reactions with ROS. This causes the destruction of molecules and the excitation of the MB molecule is relieved by internal conversion. Additionally, a different fraction of the total number of molecules converts to the excited state per unit time. Figure 5a displays the results of MTT, which demonstrate a nonlinear relationship between CECs' metabolic (proliferative) activity and both MB concentration and light exposure dose. Figure 5b,c display the results of TUNEL and TMRE assays, respectively.

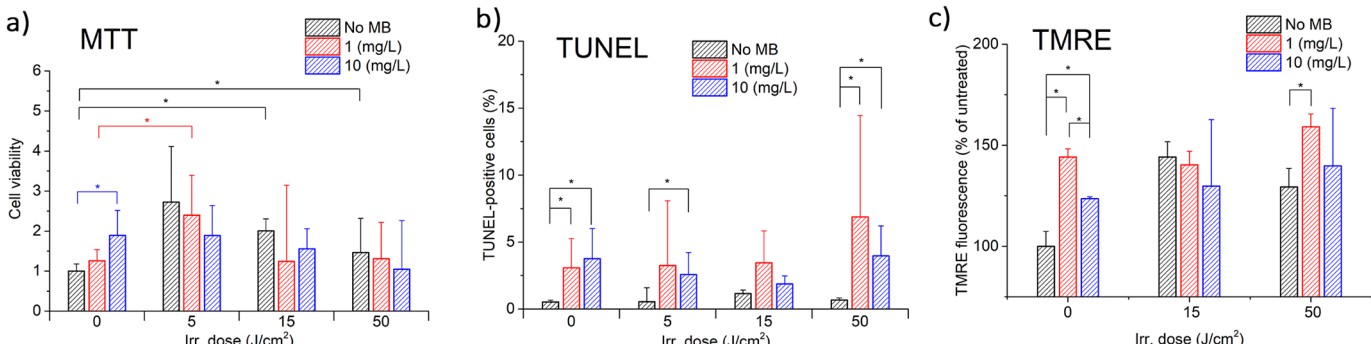

**Figure 5.** (**a**) MTT, (**b**) TUNEL, and (**c**) TMRE assays results after PDT of CECs with MB depending on MB concentration and laser dose (* $p < 0.05$, the bars show the average values, and the error bars show the standard deviation).

The MTT assay demonstrated that MB alone increased the mitochondrial activity of CECs in a concentration-dependent manner, with higher concentrations corresponding to higher metabolic activity. Laser irradiation without MB first stimulates the metabolic activity of CECs up to a light dose of 5 J/cm$^2$, resulting in a 2.7-fold increase. However, after this point, the metabolic activity decreases, showing a trend towards 100%. The reason for this effect is that photoexcitation alters the activity of cytochrome c oxidase, the primary photoacceptor molecule found in mitochondria. This alteration can result in an increase in the electrochemical proton gradient, leading to an increase in ATP synthesis, as well as other redox changes and modulation of biochemical reactions through a cascade of reactions. It is believed that these changes ultimately lead to an increase in cell proliferation [37].

Photodynamic exposure reduced the mitochondrial activity of CECs in proportion to the MB concentration. The results indicate that three competing processes are observed, depending on the MB concentration and the dose of light exposure. Further investigation is required for the energy range from 0 to 5 J/cm$^2$, which is of particular interest.

The TMRE analysis results showed an increase in mitochondrial activity. Further studies are required at lower energies ranging from 0 to 5 J/cm$^2$ to investigate the effects of MB and laser exposure on the activity of CECs mitochondria, as the differences between 15 and 50 J/cm$^2$ are not significant.

The TUNEL assay results did not indicate any significant cell death due to apoptosis. However, they are consistent with the MTT assay results, which show that higher metabolic activity corresponds to fewer apoptotic cells. The decrease in TUNEL-positive cells with 10 mg/L MB compared to 1 mg/L may be attributed to the intense damage caused by CECs. Cell-damaging factors have been shown to have paradoxical effects. A low-intensity factor induces apoptosis, while the same factor acting at high intensity induces necrosis. Intensive damage prevents cells from completing the ATP-dependent program of apoptosis. Pronounced oxidative stress suppresses the activity of redox-dependent caspases, causing the cell to switch from apoptosis to necrosis [38]. Therefore, at an MB concentration of 1 mg/L, the activity of inducing apoptosis may be more apparent. Furthermore, the early stages of apoptosis are accompanied by hyperpolarization of the mitochondrial membrane [39]. This may explain the correlation between the increase in TMRE fluorescence and the number of TUNEL-positive cells when exposed to a light dose of 50 J/cm$^2$.

It is noteworthy that, for the given MB concentrations, light doses, and laser power densities, no significant CEC death was observed. This indicates that the exposure is safe for normal CECs while offering the possibility of controlling their metabolic activity through PDT with MB.

The effect of PDT with MB on CECs was studied using fluorescence lifetime imaging microscopy (FLIM). Figure 6 shows the phasor diagrams in the spectral range of NADH and FAD before and after 50 J/cm$^2$ laser exposure with MB at a dose of 10 mg/L. The discrepancies in FAD fluorescence lifetime are most evident when the distribution over amplitude *a1* is plotted (see Figure 6, Bottom). The number of pixels where the short

component of the FAD lifetime predominates decreases 24 h after PDT (the color shifts to the red region of the palette). Consequently, the contribution of free $_f$FAD to the total fluorescence increased compared to that of bound $_b$FAD. The effect on NADH was also observed, albeit less noticeably. The contribution of bound $_b$NADH to the total fluorescence increased compared to that of free $_f$NADH 24 h after PDT. Thus, in both cases, a slight shift towards oxidative phosphorylation was observed.

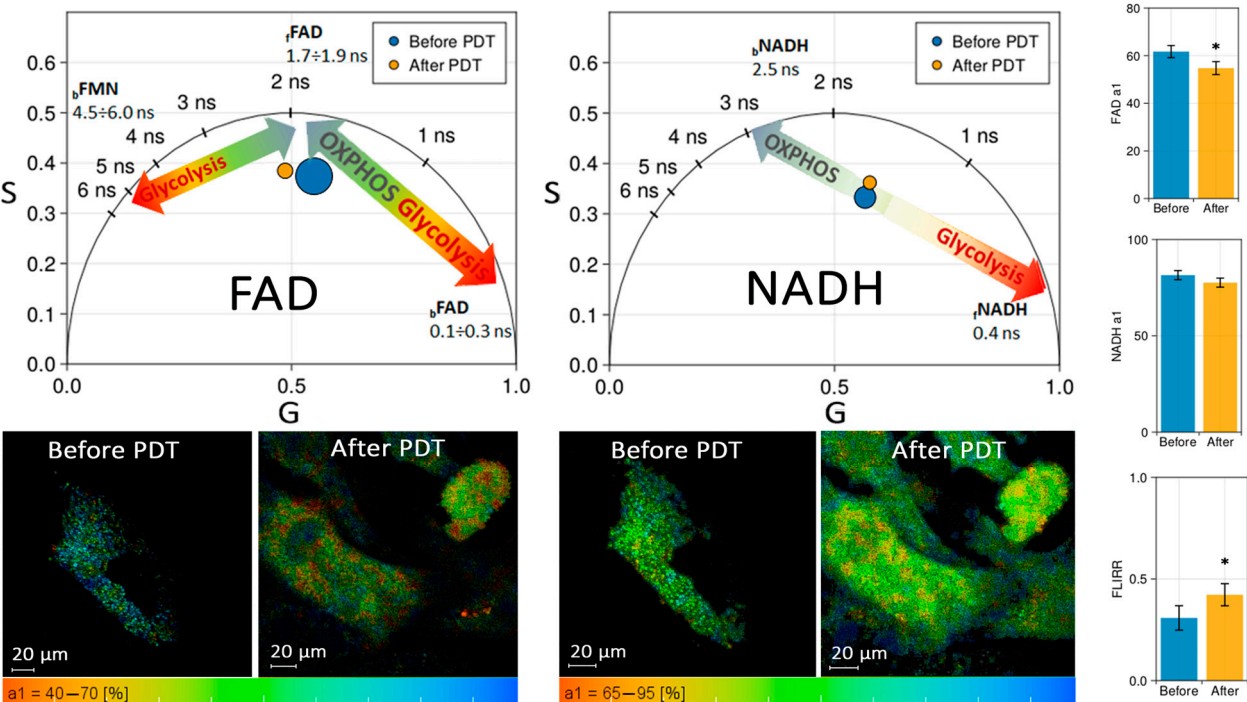

**Figure 6.** (**Top**) Phasor diagrams for time-resolved fluorescence images [40] of CECs' FAD and NADH fluorescence before and 24 h after PDT. (**Bottom**) FLIM images of CECs pseudo-colored by the distribution of the short component of fluorescence lifetime amplitude *a1*. (**Right**) Mean metabolic indexes of FAD, NADH, and FLIRR before and 24 h after PDT (* $p < 0.05$, the bars show the average values, and the error bars show the standard deviation).

FLIM results indicate that up to $50 \, \text{J/cm}^2$ of light exposure does not shift metabolism in CECs towards glycolysis, and cells maintain normal function.

## 3. Discussion

In this study, we discovered new effects of PDT and MB on CECs in vitro. In the context of brain diseases, including neurodegeneration and brain tumors, controlling angiogenesis and maintaining the permeability of the blood–brain barrier presents a significant challenge. Both mechanisms depend greatly on the metabolic status of CECs, which are characterized by an increased number and activity of mitochondria compared to microvascular endothelial cells in other tissues. Mitochondrial ATP production and glycolytic generation of lactate in CECs are both believed to contribute to the maintenance of blood–brain barrier permeability and the acquisition of different CEC phenotypes, such as tip or stalk cells, in angiogenesis [41,42]. The evaluation of controlling cerebral endothelial cells is an important task due to the role microvessels play in providing metabolic needs to transformed cells in tumor tissue. Our study shows that MB accumulates in the cytoplasm of CECs after 1 h of incubation. Figure 5 demonstrates the importance of selecting the appropriate light exposure dose based on the concentration of MB accumulated within CECs. MB is completely photobleached during PDT with a power density of $300 \, \text{mW/cm}^2$ at a light dose of $15 \, \text{J/cm}^2$.

The viability and proliferative activity of CECs change during PDT with MB, and the type of dependence is nonlinear. A concentration-dependent increase in mitochondrial activity of CECs was observed during incubation with MB. The higher the concentration of MB, the more active the mitochondria became. Secondly, laser irradiation at a wavelength of 660 nm and an energy dose of up to 5 J/cm$^2$ was found to increase mitochondrial activity by 2.7 times, followed by a decrease with a tendency to reach 100% activity. In a study by S. Taha et al. [43], similar results were found regarding the effect of laser radiation on overall cell viability.

It is important to note that the MTT results are influenced by numerous factors, such as the number and density of cells, MTT concentration, and MTT incubation period [44]. The MTT assay is an important indicator that primarily characterizes the regenerative capacity of a cell population, rather than simply reflecting the mitochondrial activity of cells. To confirm the increase in mitochondrial activity of CECs during PDT with MB, we performed TMRE analysis. The results showed a 30–60% increase in mitochondrial activity of CECs compared to the control group of non-treated cells.

Finally, a photochemical reaction caused a concentration-dependent decrease in the viability of CECs. Based on the FLIM results, it can be concluded that up to 50 J/cm$^2$ of light exposure, the metabolism of CECs remains unaffected and the cells continue to function normally. In sum, modulation of key mechanisms of energy production in mitochondria-enriched CECs with exogenous photosensitizers might be considered as a basis for the development of clinical protocols aimed to control the BBB permeability, angiogenesis, and barrierogenesis in pathological conditions associated with aberrant BBB integrity and cerebral angiogenesis (brain tumors and chronic neurodegeneration).

## 4. Materials and Methods

### 4.1. Research Design

Figure 7 presents the research design of the study.

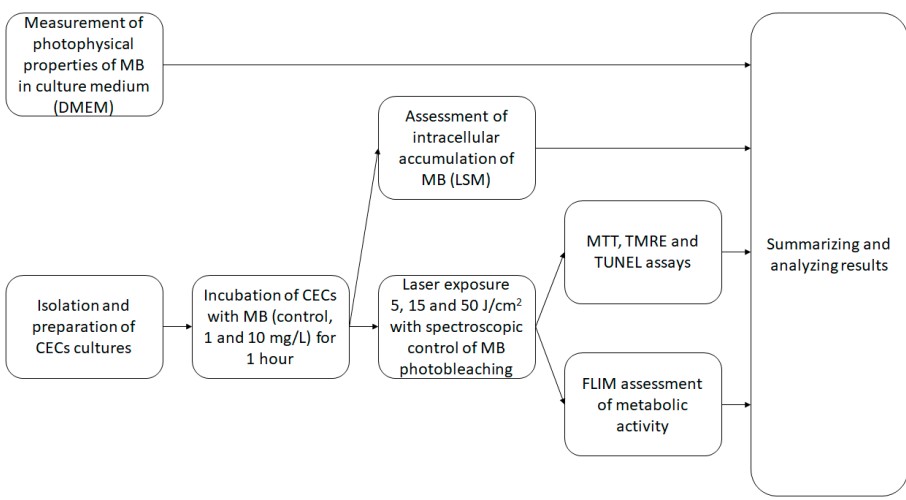

**Figure 7.** Study design.

### 4.2. Methylene Blue and Measurement of Its Photophysical Properties

Methylene blue (MB) was purchased at a pharmacy as "Methyleneblue", an aqueous solution of 1%, with the active substance methylthioninium chloride (OJSC "Samaramedprom").

A 1% aqueous solution of MB in the amount required to obtain concentrations of 1 and 10 mg/L was added to culture medium (DMEM-F12, 20% fetal calf serum, glutamine, and glucose).

The absorption spectra of samples in the ultraviolet, visible, and near-infrared (NIR) regions (325–900 nm) were measured using a U-3400 spectrophotometer (Hitachi, Tokyo,

Japan) in 1 mm thick quartz cuvettes. For data presentation, the extinction of samples was converted into optical density at 1 cm.

Fluorescence spectra were recorded using a LESA-01 spectrum analyzer (BioSpec, Moscow, Russia) with excitation by 632.8 nm CW HeNe laser (BioSpec, Moscow, Russia) in 1 cm thick quartz cuvettes. This wavelength was chosen for two reasons: it is sufficiently absorbed by the MB, and is also far enough away from the fluorescence maximum that the scattered laser radiation can be easily filtered out with a light filter.

### 4.3. Determination of Photodynamic Efficiency of MB

Photodynamic efficiency (rate of molecular oxygen consumption) in the obtained MB samples in the culture medium under laser exposure was determined by the rate of change in the hemoglobin absorption spectrum during the transition from the oxygenated to deoxygenated form. The principle of operation of this method and the experimental setup are described in detail in the work by A. Stratonnikov et al. [45]. When PS is irradiated with a laser into the absorption band, energy is transferred from PS to molecular oxygen in the system and its transition to the singlet state occurs, followed by chemical quenching, as a result of which the concentration of molecular oxygen in the system decreases. Hemoglobin is an additional source of oxygen; therefore, to restore the equilibrium concentration, hemoglobin releases stored oxygen into the system, and hemoglobin deoxygenation occurs.

To prepare samples with MB, red blood cell mass (RBC mass) diluted in culture medium (DMEM-F12, 20% fetal calf serum, glutamine, and glucose) was used. RBC mass was obtained from human venous blood. The study was conducted in accordance with the Declaration of Helsinki, and approved by the Ethics Committee of Sechenov University (protocol #22-21 dated 9 December 2021) within the framework of the Cooperation Agreement #228/20-15 576/20-4 dated 10 June 2020 between Sechenov University and NRNU MEPhI. Blood was collected in tubes with the anticoagulation agent EDTA 3K, then centrifuged at 2500 rpm for 7 min. The supernatant was removed, and RBC mass was collected and added to the culture medium in a volume ratio of 1:1. Then, 1% aqueous MB solution was added in the amount required to obtain the final MB concentrations of 1 and 10 mg/L. To record the absorption spectra of the studied samples, the LESA-01 spectrometer was used. To record the absorption spectra of hemoglobin, a broadband tungsten–halogen light source was used. A 660 nm CW semiconductor laser (BioSpec, Moscow, Russia) with fiber-optic radiation output was used as a source for excitation of MB. The power density of laser radiation on the sample surface was 100 mW/cm$^2$. The duration of irradiation was 15 min, corresponding to a dose of 90 J/cm$^2$. Due to the small volume of the sample (250 μm thickness, when recording the absorption spectrum "at the lumen"), the radiation power density of more than 100 mW/cm$^2$ can lead to significant heating of the sample, which will distort the results of the experiment. The light dose was chosen so that for MB concentration of 10 mg/L in the sample, almost all hemoglobin was converted to a deoxygenated form.

### 4.4. Cell Cultures

Cerebral vascular endothelial cells were isolated and cultured according to a modified method [46]. According to the protocol, the brain was extracted from anesthetized and sterilized 14-day-old rats; after removal of large blood vessels and white matter, the cerebral cortex was washed several times with cold Hanks' solution (HBSS), minced, and centrifuged for 5 min at 1000 rpm. A quantity of 25% bovine serum albumin was added to the precipitate and pipetted, and the homogenate was centrifuged for 10 min at 2000 rpm. The supernatant was pipetted 25–30 times and centrifuged, and 0.1% collagenase solution was added to the precipitate for 40 min at 37 °C. The fermented microvessels were centrifuged for 5 min at 1000 rpm, and the precipitate was resuspended in culture medium (DMEM-F12, 20% fetal calf serum, glutamine, and glucose) and planted on a Petri dish, changing the culture medium twice a week. The obtained cerebral endothelial cells were further transplanted into microplate wells coated with poly-l-lysine and used in the experiment when monolayer

confluency was achieved. The animals were kept in cages with free access to water and feed at a constant temperature of $21 \pm 1\,°C$ and a regular light cycle of 12 h day/12 h night. Animal studies are carried out in accordance with the principles of humanity set forth in the European Community Directive (2010/63/EC). All experiments with animals have been approved by the Local Ethic Committee (Research Center of Neurology).

### 4.5. Intracellular Accumulation

Intracellular accumulation was studied using a LSM-710-NLO laser scanning confocal microscope (Carl Zeiss, Oberkochen, Germany). A 32-channel GaAsP detector was used to record fluorescence spectra. A 633 nm laser was used to excite MB fluorescence and 488 nm for autofluorescence.

A LESA-01-BIOSPEC fiber-optic spectrometer (BIOSPEC, Moscow, Russia) was used to record the integrated fluorescence of MB inside the cells in the plate. A 632.8 nm HeNe laser was used for excitation. Five fluorescence spectra were measured in each well, after which the results were averaged and the area under the fluorescence peak and standard deviation were calculated (Figure 4b,d).

### 4.6. Photodynamic Treatment

The incubation time of MB in a 96-well plate at concentrations of 1 and 10 mg/L (3 and 30 μM respectively) before irradiation was 1 h. After incubation with MB, cells were washed with PBS 3 times and then immersed in fresh culture medium. Cells were irradiated at the following doses—5, 15, and 50 J/cm². Irradiation was performed with a 660 nm CW semiconductor laser (BioSpec, Moscow, Russia) with fiber-optic radiation output at a power density of 300 mW/cm² (Figure 8). Such parameters of laser radiation did not lead to heating of the sample by more than 3 °C. There were wells exposed to irradiation, but to which MB was not added in any concentrations. Intact cells without MB and without irradiation served as controls. Four wells of a 96-well plate were used for each MB concentration and each energy dose in MTT, TMRE, and TUNEL assays.

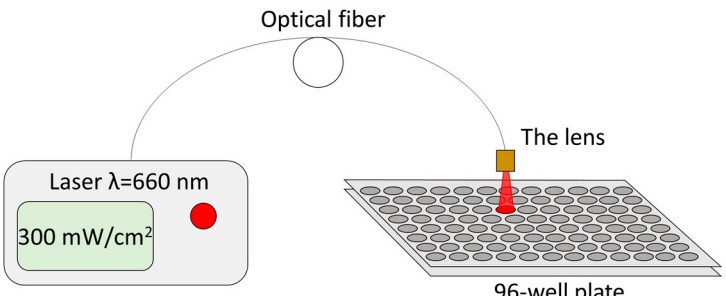

**Figure 8.** The scheme of the experimental installation for performing PDT of the CECs with MB.

### 4.7. Estimation of CEC Viability

To assess cytotoxic activity, cells were incubated for 24 h under darkened conditions in a $CO_2$ incubator. Unexposed cells were used as controls. Cell survival was assessed using a colorimetric MTT assay. Before the test, the incubation medium in the wells of the plate was replaced with fresh medium (200 μL in each well). Then a pre-prepared 0.5% solution of 3-(4,5-dimethylthiazolyl-2)-2,5 diphenyltetrazolium bromide was added to the microplate wells 20 μL at a time and incubated for 2 h under standard $CO_2$ incubator conditions. After completion of incubation, the MTT reagent medium was removed and 100 μL of dimethyl sulfoxide (DMSO) was added to the wells. Optical density was measured on a Multiscan Multichannel Tablet Photometer (Thermo Fisher Scientific, Waltham, MA, USA) at a wavelength of 550 nm. Cell viability in culture was calculated according to the formula:

$$CV = 100 \times (OD_e - OD_m)/(OD_c - OD_m),$$

where $OD_e$—optical density of formazan in experimental wells, $OD_c$—optical density of formazan in control wells, $OD_m$—optical density of medium.

### 4.8. Assessment of Mitochondrial Activity (TMRE Assay)

TMRE selectively accumulates in active mitochondria due to the transmembrane mitochondrial potential, which allows the assessment of mitochondrial activity in cell culture. The final concentration of TMRE in each well is 200 nM. The culture with dye was incubated in $CO_2$ incubator for 15 min, then the wells were washed three times with modified Ringer–Locke's solution. The EVOS M7000 cell visualization system (Thermo Fisher Scientific, USA) was used to assess the fluorescence intensity of endotheliocyte mitochondria.

### 4.9. CECs Apoptosis (TUNEL Assay)

Cell apoptosis was determined 24 h after exposure in a 96-well plate using the TUNEL in situ kit (Elabscience, Houston, TX, USA). For this purpose, cells were fixed with 5% chilled formalin solution, then washed twice for 5 min in PBS, and 100 μL of $1\times$ Protein Kinase K working solution was added from the kit to the well and incubated for 10 min at 37 °C, then washed with PBS 3 times for 5 min each.

The next step was to add 100 μL of DNase $1\times$ Buffer from the kit to each well, followed by incubation for 30 min at 37 °C and washing with PBS 3 times for 5 min each. Next, 100 μL of TdT Equilibration Buffer was added with analogous incubation conditions, and 50 μL of working solution consisting of 35 μL of TdT Equilibration Buffer, 10 μL of labeling solution, and 5 μL of TdT enzyme was added and incubated for one hour at 37 °C. After washing 3 times with PBS for 5 min each, cells were stained with DAPI solution for 5 min and analyzed on a fluorescence microscope using a TRITC filter.

Digital images were acquired using an EVOS M7000 imaging system (Thermo Fisher Scientific, USA) and processed in ImageJ 1.54p software using a plugin for counting fluorescent marks on microphotographs.

To assess the level of apoptosis, the total number of TUNEL-positive cells normalized to 100 DAPI-positive cells was counted, which is presented as the mean number with standard deviation.

### 4.10. Metabolic Activity of CECs

Spectra and time-resolved images of autofluorescence were recorded under two-photon excitation with a Cameleon Ultra II femtosecond laser (Coherent, Saxonburg, PA, USA) at a wavelength of 740 nm. The fluorescence lifetime of NADH and FAD coenzymes in cells before PDT and 24 h after this exposure was recorded using a FLIM module (Becker&Hickl, Berlin, Germany) attached to LSM-710-NLO, consisting of a time-correlated photon counting system (TCSPC) SPC-150 and hybrid GaAsP photodetector HPM-100-07, SPC-IMAGE 8.8 software. Bandpass filters upstream of the time-resolved detector FB450-40 (Thorlabs, Newton, MA, USA) and FB550-40 (Thorlabs, Newton, MA, USA) were used to isolate fluorescence signals from NADH and FAD, respectively. Time-resolved fluorescence images were processed using SpcImage 8.0 software (Becker&Hickl, Berlin, Germany). To interpret the time-resolved fluorescence from different spectral ranges corresponding to NADH and FAD, a phasor diagram approach was applied, where the fluorescence lifetime is stated in a frequency representation [47]. KernelDensity and Makie libraries of the Julia programming language were used to construct the phasor diagrams. The phasor diagram represents the fluorescence decay curve in terms of phase parameters G (x axis) and S (y axis)—the real and imaginary parts of the first element in the Fourier series of the repeating decay signal. The distributions were obtained using kernel-density estimation weighted by pixel intensity. The semicircle on the phasor diagram represents the values for mono-exponential decay. The points inside the semicircle are a superposition of several lifetimes, and the points outside the circle represent sub-exponential decay possibly due to the long lifetime decay curve overlap.

In calculating the NADH $a_1/a_2$ metabolic index, kinetics were approximated using fixed lifetimes: NADH $\tau_1$ = 0.4 ns, $\tau_2$ = 2.5 ns; FAD $\tau_1$ = 0.25 ns, $\tau_2$ = 1.4 ns and $\tau_3$ = 5.0 ns [48]. The NADH fluorescence lifetime curve fitting uses two components. The FAD fluorescence lifetime curve fitting uses three components. For each cell sample, 2 images were taken before and after PDT. The average number of cells per frame was 10. Metabolic indices were calculated for individual cells, then the results were averaged and the standard deviation calculated.

### 4.11. Statistical Analysis

Four wells of a 96-well plate were used for each MB concentration and each energy dose in MTT, TMRE, and TUNEL assays. The Mann–Whitney U-test was used for statistical processing of the data using the program StathisTisa 7 (StatSoft Inc., Moscow, Russia). The results were considered significant at $p < 0.05$. Data are presented as mean and standard deviation.

**Author Contributions:** Conceptualization, V.I.M., A.B.S. and V.B.L.; methodology, V.I.M., A.S.S. and A.S.A.; software, V.I.M.; validation, V.B.L.; formal analysis, V.I.M., A.B.S. and V.B.L.; investigation, V.I.M., A.S.S., A.S.A., A.K.B. and M.V.C.; data curation V.I.M.; writing—original draft preparation, V.I.M.; writing—review and editing, A.S.S., V.B.L., A.S.A., A.K.B., M.V.C. and A.B.S.; visualization, V.I.M.; supervision, A.B.S., V.B.L.; project administration, V.I.M. All authors have read and agreed to the published version of the manuscript.

**Funding:** Spectroscopic, microscopic and photodynamic studies were funded by a grant from the Russian Science Foundation (project N 22-72-10117). Studies on isolation, culture, assessment of viability and mitochondrial function in brain endothelial cells were funded by the Government Assignment for Research, Ministry of Science and Higher Education of the Russian Federation, «Aberrant metabolic plasticity of cells within the neurovascular unit in brain pathology» (1023101100004-9-3.1.8;3.1.4., Research Center of Neurology).

**Institutional Review Board Statement:** Not applicable.

**Informed Consent Statement:** Not applicable.

**Data Availability Statement:** The raw data supporting the conclusions of this article will be made available by the authors on request.

**Acknowledgments:** The team of authors would like to thank Igor D. Romanishkin, Daria V. Pominova, and Anastasia V. Ryabova from the Prokhorov General Physics Institute of the Russian Academy of Sciences for help in processing the results of the experiment.

**Conflicts of Interest:** The authors declare no conflicts of interest.

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
