# Peer review of "Effect of Photodynamic Therapy with the Photosensitizer Methylene Blue on Cerebral Endotheliocytes In Vitro"

_photonics, doi:10.3390/photonics11040316_

Round 1

Reviewer 1 Report

Comments and Suggestions for Authors

The manuscript of Makarov et al. is focused on some aspects of dosimetry in characterisation of the photodynamic effect with the methylene blue on cerebral endothelial cells. In general, the logical structure of the text is appropriate, but could be improved. The manuscript should be carefully checked for linguistic, editorial, and typo errors. Furthermore, clarification, inclusion, or correction of certain information in the manuscript may be required.

 The major comments:

 -The manuscript lacks information on sample size, how many samples, how many measurements were made, how many images were taken, how many absorption/fluorescence spectra were acquired - without such information it is difficult to determine the statistical significance of the results obtained.

-Lines 187-188: “RBC mass was 187 obtained from human venous blood.” – No information about the approval number for studies involving humans according to the guidelines of the Declaration of Helsinki. I do not understand why the authors did not consider this issue at all ('Institutional Review Board Statement: Not applicable').

-Lines 200-201: “the brain was extracted from anesthetized and sterilized 14-day-old rats; after removal of large blood vessels and white matter” - – no information about the approval number for studies involving animals according to the guidelines of the Declaration of Helsinki. I do not understand why the authors did not consider this issue at all ('Institutional Review Board Statement: Not applicable').

-As stated by Authors “Irradiation was performed with a 660 nm laser at a power density of 300 mW/cm2 .” (lines 224-225) and “The power density of 195 laser radiation on the sample surface was 100 mW/cm2” (lines 195-196)- the power densities are quite high, but there is no information in the manuscript as to whether thermal effects on the cells were also investigated.

-Lines 304-305: “However, for MB concentration of 10 mg/L, a second intense absorption peak was observed in the red region of the spectrum at 600-630 nm,” I would say that 2 additional maxima/peaks appear in this spectral range for such a high concentration of MB.

-Fig. 2 A – I do not understand the absorption spectra. Why initially the high absorption maximum of DMEM at wavelength 556 nm decreased for a lower concentration (1mg/L) of MB, and then increased for a higher MB concentration (10mg/L)? These spectra were averaged? What was the sample size?

-For such a high concentration of MB (10 mg/L) no aggregates were observed?

-Line 306: “Presence of a high concentration of dimers in this sample” – Dimers are contributors to MB fluorescence?

-Figure 3 – Please indicate in the figure caption the detailed photoexcitation and exposure parameters.

-Unsupported statement:We do not have a strict proof of the proposed mechanism, this effect requires further investigation, and it was not the main purpose of this work. However, in our opinion, the mechanism described above is the most probable.'Any references?

-Fig. 4 and 'A marked decrease in the intensity of MB fluorescence in CECs was observed under laser exposure 357 due to photobleaching.'why do the authors think that this effect is related only to photobleaching? Do the authors have experimental confirmation that this effect has nothing to do with the ROS generation process?

-In a rather lengthy introduction, the authors do not describe very clearly what their results actually add to the state of knowledge.

-The description of the methodology is quite complex (11 subsections of Section 2). The authors should consider including some kind of diagram to present the workflow in a more accessible way.

-The abstract is too wordy. It should be more concise and contain the most important information. According to the journal's requirements, the abstract should be no longer than 200 words.

Minor comments:

The manuscript should be carefully checked for linguistic, editorial, and typographical errors (eg.  Line 29: 'doses of - 5, 15 and 50 J/cm2” - a colon would be better or nothing at all; line 200: '... method [39] According' - no full stop at the end of the sentence; Fig.5: the Figure and caption should be on the same page.)

Author Response

The document contains the comments and questions of the reviewers  and our responses to them (italics).

The authors express their deep gratitude to all reviewers for the work done!
The places where corrections were made can be seen in the attached "marked manuscript" document

The manuscript of Makarov et al. is focused on some aspects of dosimetry in characterisation of the photodynamic effect with the methylene blue on cerebral endothelial cells. In general, the logical structure of the text is appropriate, but could be improved. The manuscript should be carefully checked for linguistic, editorial, and typo errors. Furthermore, clarification, inclusion, or correction of certain information in the manuscript may be required.

 The major comments:

 -The manuscript lacks information on sample size, how many samples, how many measurements were made, how many images were taken, how many absorption/fluorescence spectra were acquired - without such information it is difficult to determine the statistical significance of the results obtained.

Thank you for your valuable comment! We have added to the chapter Materials and Methods paras. 4.6, 4.10 и 4.11.

-Lines 187-188: “RBC mass was 187 obtained from human venous blood.” – No information about the approval number for studies involving humans according to the guidelines of the Declaration of Helsinki. I do not understand why the authors did not consider this issue at all ('Institutional Review Board Statement: Not applicable').

We agree with the comment! We have added information about the permission to 4.3.

-Lines 200-201: “the brain was extracted from anesthetized and sterilized 14-day-old rats; after removal of large blood vessels and white matter” - – no information about the approval number for studies involving animals according to the guidelines of the Declaration of Helsinki. I do not understand why the authors did not consider this issue at all ('Institutional Review Board Statement: Not applicable').

Information of approval for research involving animals has been added to 4.4.

-As stated by Authors “Irradiation was performed with a 660 nm laser at a power density of 300 mW/cm2 .” (lines 224-225) and “The power density of 195 laser radiation on the sample surface was 100 mW/cm2” (lines 195-196)- the power densities are quite high, but there is no information in the manuscript as to whether thermal effects on the cells were also investigated.

Temperature was not controlled in this study. However, all our previous studies with heating control showed that at such concentrations and power densities the temperature change did not exceed 3 °C both in vitro and in vivo.

-Lines 304-305: “However, for MB concentration of 10 mg/L, a second intense absorption peak was observed in the red region of the spectrum at 600-630 nm,” I would say that 2 additional maxima/peaks appear in this spectral range for such a high concentration of MB.

The main absorption peak of MB is near 660-665 nm (red arrow), and an additional peak occurs when MB molecules dimerize at high concentrations (blue arrow) (see attached file).

-Fig. 2 A – I do not understand the absorption spectra. Why initially the high absorption maximum of DMEM at wavelength 556 nm decreased for a lower concentration (1mg/L) of MB, and then increased for a higher MB concentration (10mg/L)? These spectra were averaged? What was the sample size?

It was important for us to show the fundamental shape of the absorption spectrum of the studied samples, so the concentration of DMEM in the measured samples could differ due to their preparation method - different volumes of aqueous MB solution were added to the 1 and 10 mg/L samples, so the final concentration of DMEM in the 0, 1, and 10 mg/L samples differed.  Therefore, a difference in absorbance at 556 nm wavelength was obtained. We have taken your comment into account and added an explanation to Figure 2a.

Since these measurements were performed without adding cells to the sample, it is in our opinion inappropriate to talk about sample size. Differences from sample to sample with the same concentrations will be determined by the error of the measuring instrument.

-For such a high concentration of MB (10 mg/L) no aggregates were observed?

According to the literature [Fernandez-Perez, Amparo, and Gregorio Marban. "Visible light spectroscopic analysis of methylene blue in water; what comes after dimer?". ACS omega 5.46 (2020): 29801-29815.], the absorption peak of the aggregates is broader than that of the dimer and its maximum is near 600 nm. According to our absorption spectra, monomeric and dimeric forms should have been present in the samples.

-Line 306: “Presence of a high concentration of dimers in this sample” – Dimers are contributors to MB fluorescence?

No, due to self-extinction, the fluorescence of MB dimers is negligible compared to the monomeric form. Morgounova, Ekaterina, et al. "Photoacoustic lifetime contrast between methylene blue monomers and self-quenched dimers as a model for dual-labeled activatable probes." Journal of biomedical optics 18.5 (2013): 056004-056004.

-Figure 3 – Please indicate in the figure caption the detailed photoexcitation and exposure parameters.

Thanks for the comment! We have added photoexposure and exposure parameters to the text in the caption of Figure 3.

-Unsupported statement: “We do not have a strict proof of the proposed mechanism, this effect requires further investigation, and it was not the main purpose of this work. However, in our opinion, the mechanism described above is the most probable.' – Any references?

Unfortunately, we did not find direct confirmation of our hypothesis in the literature; however, our research group is currently investigating this effect. Indirect evidence of the proposed mechanism is given in Moreira, Leonardo Marmo, et al. "The methylene blue self-aggregation in water/organic solvent mixtures: Relationship between solvatochromic properties and singlet oxygen production." Orbital: The Electronic Journal of Chemistry (2017): 279-289.

-Fig. 4 and 'A marked decrease in the intensity of MB fluorescence in CECs was observed under laser exposure 357 due to photobleaching.' – why do the authors think that this effect is related only to photobleaching? Do the authors have experimental confirmation that this effect has nothing to do with the ROS generation process?

Thank you for the comment! By photobleaching we understand a complex process associated with many effects. These include both direct photodissociation of MB molecules and destruction of MB molecules mediated by AFC and other specific reactions. We have added an explanation of this fact to the text of the article, as well as a reference to the study of the mechanisms of MB photobleaching [Nassar, Sulafa Jamal M., Corinne Wills, and Anthony Harriman. "Inhibition of the photobleaching of methylene blue by association with urea." ChemPhotoChem 3.10 (2019): 1042-1049].

-In a rather lengthy introduction, the authors do not describe very clearly what their results actually add to the state of knowledge.

We agree with the comment and have added the scientific novelty of the study to the text of the Introduction of the article

«Achieving this goal, namely, the modulation of key energy production mechanisms in mitochondrial-enriched CECs by exogenous photosensitizers, can be considered as the basis for the development of clinical protocols aimed at controlling BBB permeability, angiogenesis and barrier genesis in pathological conditions associated with impaired BBB integrity and cerebral angiogenesis, for example, brain tumors or chronic neurodegeneration».

-The description of the methodology is quite complex (11 subsections of Section 2). The authors should consider including some kind of diagram to present the workflow in a more accessible way.

Thanks for the comment, we have added the design/plan of the experiment to the Materials and Methods chapter 4.1.           

-The abstract is too wordy. It should be more concise and contain the most important information. According to the journal's requirements, the abstract should be no longer than 200 words.

We have shortened the Abstract considerably

Minor comments:

The manuscript should be carefully checked for linguistic, editorial, and typographical errors (eg.  Line 29: 'doses of - 5, 15 and 50 J/cm2” - a colon would be better or nothing at all; line 200: '... method [39] According' - no full stop at the end of the sentence; Fig.5: the Figure and caption should be on the same page.)

The comments have been corrected 

Reviewer 2 Report

Comments and Suggestions for Authors

Overall, the manuscript appears to be well-written. In addition, I think that the manuscript might deserve publication in  photonics after some points are dealt with and some missing details are added prior to publication as follows:

1-      Please include in the text more details about the laser source used for PDT. Also, please specify whether it is CW or pulsed laser light.

2-      The authors should explain what makes this work unique and how it differs from previous research. 

3-      Please explain in the text why the excitation source for Methylene blue (MB) was a 632.8 nm CW He-Ne laser.

4-      Please include in the text a brief sketch that describes the PDT experiment.

5-      Please discuss in the text why the excitation parameters as 100 mW/cm2 power density,  15 minutes duration of irradiation, and dose of 90 J/cm2 were selected to record the absorption spectra of hemoglobin.

6-      Please discuss in the text why the irradiation parameters as 660 nm laser wavelength and power density of 300 mW/cm2 were selected for the Photodynamic treatment.

7-      Please describe the safety and biocompatibility of photosensitizer methylene blue and laser treatment on cerebral endothelial cells.

8-      Nothing is mentioned in the text about the uncertainty in the experimental results, so please discuss this issue in the text.

9-      In addition to the References list, other current research on the use of laser light as a PDT tool is proposed to be reviewed and included if it is beneficial:

Ø    Journal of photochemistry and photobiology. B, Biology vol. 240 (2023): 112665.

                      doi:10.1016/j.jphotobiol.2023.112665

Comments on the Quality of English Language

Minor editing of English language required

Author Response

The document contains the comments and questions of the reviewers  and our responses to them (italics).

The authors express their deep gratitude to all reviewers for the work done!
The places where corrections were made can be seen in the attached "marked manuscript" document

Overall, the manuscript appears to be well-written. In addition, I think that the manuscript might deserve publication in  photonics after some points are dealt with and some missing details are added prior to publication as follows:

1-      Please include in the text more details about the laser source used for PDT. Also, please specify whether it is CW or pulsed laser light.

Thank you for the comment! We have added information about the laser source used for PDT in 4.3 and 4.6

2-      The authors should explain what makes this work unique and how it differs from previous research. 

We agree with the comment and have added a description of the novelty of the work to the last paragraph of the Introduction.

3-      Please explain in the text why the excitation source for Methylene blue (MB) was a 632.8 nm CW He-Ne laser.

An explanation of the reason for source selection has been added in Section 4.2.

4-      Please include in the text a brief sketch that describes the PDT experiment.

We have added a graphical representation of the PDT scheme in p.4.6

5-      Please discuss in the text why the excitation parameters as 100 mW/cm2 power density,  15 minutes duration of irradiation, and dose of 90 J/cm2 were selected to record the absorption spectra of hemoglobin.

We added in Section 4.3 the reasons for selecting the excitation parameters to be used.

6-      Please discuss in the text why the irradiation parameters as 660 nm laser wavelength and power density of 300 mW/cm2 were selected for the Photodynamic treatment.

According to our preliminary studies, such a power density and at such MB concentrations leads to heating of the sample by less than 3 degrees. Since the wells were irradiated alternately, it was necessary to minimize the total exposure time so that MTT, TMRE, and TUNEL assays were performed under the same conditions for cells from each well. We have also added an explanation in section 4.6.

7-      Please describe the safety and biocompatibility of photosensitizer methylene blue and laser treatment on cerebral endothelial cells.

The issue of safety and biocompatibility of MB PS and laser treatment of CECs has been covered in the Introduction.

8-      Nothing is mentioned in the text about the uncertainty in the experimental results, so please discuss this issue in the text.

We have added information about sample size, number of samples, number of measurements in 4.6, 4.10 and 4.11. We also added explanations about mean values and standard deviation in the figure captions.

9-      In addition to the References list, other current research on the use of laser light as a PDT tool is proposed to be reviewed and included if it is beneficial:

Ø    Journal of photochemistry and photobiology. B, Biology vol. 240 (2023): 112665.

                      doi:10.1016/j.jphotobiol.2023.112665

Thank you for your valuable suggestion, we have compared our results with the results from this article and included this study in the reference list

Round 2

Reviewer 1 Report

Comments and Suggestions for Authors

In my opinion, the manuscript in its current form is acceptable for publication.

Author Response

The authors are very grateful to the reviewer for his valuable comments and work!

Reviewer 2 Report

Comments and Suggestions for Authors

Overall, the manuscript looks to have been written clearly and thoroughly. The writers made sufficient changes to the paper in response to my previous suggestions and concerns. In summary, the manuscript reads well and helps to clarify the authors' manuscript. In my opinion, the manuscript now has all of the information and is ready for publishing in Photonics.

Comments on the Quality of English Language

Moderate editing of English language required

Author Response

The authors are very grateful to the reviewer for his valuable comments and work!

We have made every effort to improve the editing of English language in the article. The corrections can be seen in the attached file "Marked manuscript".
